# How effective is TabStructNet in capturing the structure of a table-image into an XML? : A reproducibility report

## Reproducibility Summary

### Scope of Reproducibility

In this submission, findings when attempting to reproduce the results from the EECV 2020 article (3) by using the corresponding code-repository github.com/sachinraja13/TabStructNet (made available by the authors of (3), i.e., by Raja et al.) are reported. Each challenge encountered, along with the corresponding solution – which was either discovered or was learnt from the first author of (3) himself – is described step-by-step. As a consequence, the intermediate files that one would manage to (and one needs to) generate at those steps, along with their inter-relationships with the rest of the code-repository have also been detailed. A few recommendations are put forward in process which might help the authors to make the repository more consistent with the paper, user-friendly, and as a consequence, to make the experiments more easily reproducible. In this submission, a few minor deviations of the model architecture from what is described in (3) to what is observed in the TabStructNet code repository are also reported. It is hoped that this report will make it easier for everyone to use and/or rebuild the described TabStructNet model.

### Methodology

Evaluated the model proposed by Raja et al. (3), using the code repository they made available at TabStructNet on own LaTeX-generated table-images.

### Results

The TabStructNet model was found to be quite good at achieving precise detection of all of the cells present in a table. The evaluation tests reported herein were done on a few table-images generated by the author of this paper himself, which therefore, the TabStructNet pretrained model has never seen.

### What was easy

Reading and understanding both the paper and the code implementation was easy. Both the writing and the scripting by the authors of (3) were clear and concise, which made it easier to find the missing blocks, a few inconsistencies between the paper and the implementation.

### What was difficult

Some of the codeblocks, particularly JSON and XML generation modules, were missing from the code-repository. Also, the code repository structure also had to be modified a little for the predictions to work. Particularly, 'TabStruct-Net/mrcnn' folder needs to be moved to 'TabStructNet/samples/tab/.' JSON and XML file generation also necessitates explicit movement of the intermediate files by the user currently, going by the advice from the first author of (3).

### Communication with original authors

Upon hitting a roadblock in running the scripts in the provided repository on own test-images (detailed in this report, primarily due to missing JSON and XML generation modules), authors of (3) were contracted. While there was some delay in getting the initial response (20 days), the first author of (3) addressed the stated queries by email quite professionally, providing the missing modules.

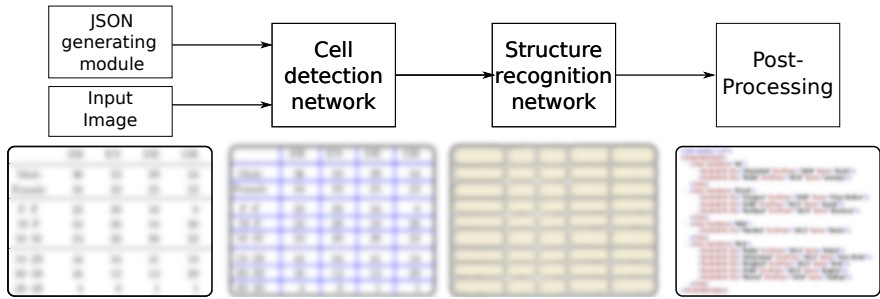

Figure 1: Table-image to XML-based table generation process pipeline.

## 1 Introduction

Tables are effective at summarising and communicating a complex information through only the precise and necessary data – getting rid of the otherwise grammar- and language-induced verbosity. They are, thus, ubiquitous; especially in the finance and science sectors, e.g., we find them in invoices, tax- and bank-statements, medical records, equipment- and facility-related logs. Owing to the multitudes of possibilities that exist for a table template in terms of various foreground and background colours, font sizes and font types, presence and absence of various vertical and horizontal line types, widths and designs, possible existence of multicolumn or multiline cells, i. e., vertical and horizontal cell merges, various levels of column- and row-spacings, different vertical and horizontal text alignments possible. Machine-understanding and regeneration of the scanned hand-written and printed tables is where arguably lies the core and a battlefield for the multi-million document analysis industry. This is not only because the table information extraction is challenging, but also because the task is highly demanding in terms of the accuracy and precision requirements, thanks to the criticality of the data that the tables typically represent . Importance of this task is quite evident from the fact that at least one special session on this particular topic is held in almost every International Conference on Document Analysis and Recognition (ICDAR). Two separate table information extraction challenges have been organized as part of the ICDAR 2021 as well.

### 1.1 Table-image to XML generation pipeline from (3), and the deviations discovered

The authors of (3) propose an end-to-end system to recognise the structure of a table present in any given image, to ultimately generate an XML containing that predicted strucutre in terms of the bounding boxes, spanning information, and the cell contents. A redrawn version of the XML table generation pipeline is presented in Figure 1. The authors describe the process of generating XML from a table image by splitting it into three distinct components, namely the 'Cell detection network', 'Structure recognition network' and the 'Post-processing' module that generates the XML output (3).

**Deviation 1: The crucial 'Post-processing' XML-generating module was missing from the TabStructNet.**

Looking at the TabStructNet repository, one can see that it did not feature the crucial 'Post-processing' module initially.

**Deviation 2: The necessary JSON generation module missing from the TabStructNet repository, also not described in (3).**

Evaluating the TabStructNet model on any test image necessitates that a JSON file with mock labels be provided as an input to the model, generation block of which is not included in the repository.

As we would see later in Section 2, failure to provide this JSON file results into an early termination of the program with an error. The first author of (3) kindly responded by email and provided the zipped directory structures for both the script-modules, along with the detailed instructions. However, the two components are still not part of the repository, which is currently the biggest limitation of the provided code-base.

It is an easily avoidable, yet a severe stumbling block of the repository invariably leading user to an error message. Because this problem – while a major one leading one to an early script termination – can be easily avoided, the reproducibility of the claims and the results presented in (3) cannot be challenged on the basis of this issue alone. Only upon taking care of the necessary dependencies manually, one can make any strong claims regarding the reproducibility and capability of the TabStructNet presented in (3). In order to make sure that the model works on any real-life test data, i. e., there is no hack (e. g., remembering and recording the instances labels provided in the repository as part of the h5 file with further obfuscation), one needs to test the performance of the model on unseen test images. The test

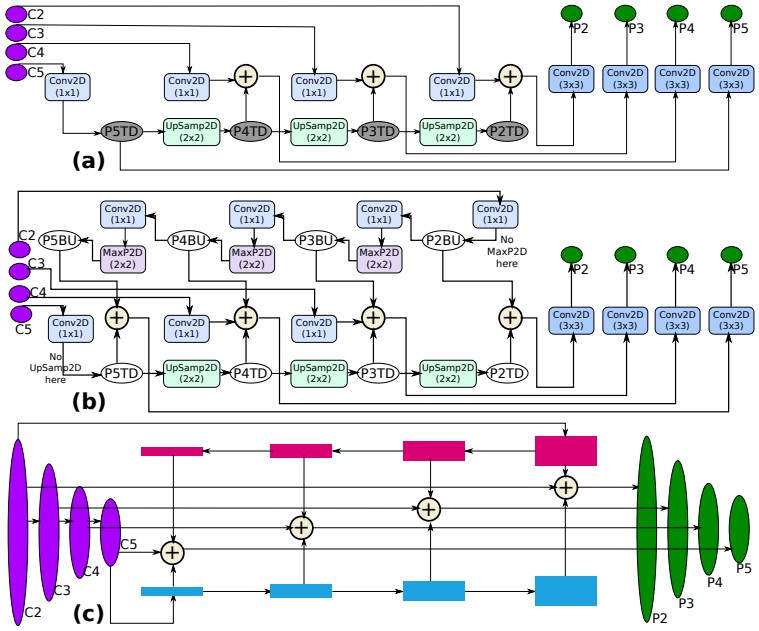

Figure 2: (a) The Feature pyramid network (FPN) from (2, 1) as implemented in the Matterport's Mask_RCNN repository, modifying which the FPN present in TabStructNet is built. We note that the computational graphs for $P2$, $P3$ and $P4$ are similar. While the tensor placeholders $P\{N\}TD$ do not even exist in Matterport's Mask_RCNN, these have been marked in gray above at equivalent places, to help simplify the comparison between these two FPN architectures.

(b) The FPN from TabStructNet (3) that includes both the 'Bottom Up' and 'Top Down' pathways. Notice that, while $P3$ and $P4$ computation graphs are similar (i. e., a summation of 3 inputs, followed by a 2-d convolution), $P2$ and $P5$ computation graphs are both different, featuring a summation operation over only two inputs. This, of course, is not a criticism of the architecture. We only note the perceived contradictions with respect to what the Figure 5 from (3) leads one to believe. (c) Redrawn Figure 5 from (3) for an easy comparison.

74 images could come from private resources that Raja et al. have no access to. We should ideally generate these ourselves
75 for model testing, so that we are even more sure that the model is presented with an image it has never seen.

**Deviation 3:** **Feature pyramid network (FPN) implementation is different than what it appears to be from**
77 **Figure 5 of (3).**

78 The key difference between the FPN from the Matterport Mask-RNN and that from the TabStructNet pretrained model is
79 the newly introduced bottom-up pathway in the FPN of TabStructNet. Notice from Figure 2(b) that the graph structures
80 in the top-down and bottom-up pathways are different for the $\{C2, C3, C4, C5\}$ to $\{P2, P3, P4, P5\}$ computations.
81 Specifically, for $N = \{3, 4\}$, $P\{N\}$ tensors are results of a 2-d convolution operation over a summation of three
82 tensors, $Conv2D(C\{N\})$, $P\{N\}TD$ and $P\{N\}BU$. However, for $N = \{2, 5\}$, $P\{N\}$ tensors are results of a 2-d
83 convolution operation over a summation of only two tensors each. Formally,

$$
\begin{aligned}
P2 =\ & Conv2D(Conv2D(C2)(:= P2BU) + P2TD), \\
P3 =\ & Conv2D(Conv2D(C3) \qquad\quad + P3TD + P3BU), \\
P4 =\ & Conv2D(Conv2D(C4) \qquad\quad + P4TD + P4BU), \\
P5 =\ & Conv2D(Conv2D(C5)(:= P5TD) + P5BU),
\end{aligned}
$$

$$
\begin{aligned}
\text{where, } P2BU =\ & Conv2D(\qquad\ C2 \qquad\quad), P5TD = Conv2D(C5), \\
P3BU =\ & MaxPool2D(\quad Conv2D(P2BU) \quad), P4TD = UpSample2D(P5TD), \\
P4BU =\ & MaxPool2D(\quad Conv2D(P3BU) \quad), P3TD = UpSample2D(P4TD), \\
P5BU =\ & MaxPool2D(\quad Conv2D(P4BU) \quad), P2TD = UpSample2D(P3TD).
\end{aligned}
$$

84 The inconsistency reported here is merely a result of a somewhat incorrect illustration describing a TabStructNet
85 component. This in itself does not pose a serious concern or suspicion in terms of reproducibility of the model, or its
86 effectiveness. The correction is presented for the sake of completeness, and as a quick caveat in the interest of those
87 looking to rebuild the model from scratch.

## 2 Setting up to evaluate TabStructNet: The challenges and the solutions

### 2.1 Make sure that the mrcnn package is NOT installed

If one has some mrcnn package preinstalled in the python working environment, running 'samples/tabnet/tabnet.py' would instantiate the installed mrcnn module components and not the custom mrcnn modules provided in the TabStructNet repository. While one is still able to load the model and the model weights, the script 'tabnet.py' tries to access r["row_adj"] where r is the model detect output. Because the default mrcnn package as part of the Python Packaging Authority (PyPA) does not return detections that featuring "row_adj" and "col_adj" as keys, the evaluation would terminate with a KeyError.

```
Running TAB evaluation on 1 images.
Traceback (most recent call last):
File "samples/tabnet/tabnet.py", line 624, in <module>
limit=int(args.limit))
File "samples/tabnet/tabnet.py", line 394, in evaluate_tabnet
row_adj = r["row_adj"]
KeyError: 'row_adj'
```

### 2.2 Make sure to generate a JSON file at '/trained_model/tab/annotations/'

If you do not provide a JSON file, you run into the following error

```
loading annotations into memory...
Traceback (most recent call last):
File "samples/tabnet/tabnet.py", line 575, in <module>
return_tab=True,
File "samples/tabnet/tabnet.py", line 101, in load_tab
dataset_dir, subset, year))
File "/home/<username>/anaconda2/envs/tf1/lib/python3.7/site-packages/
    pycocotools/coco.py", line 84, in __init__
with open(annotation_file, 'r') as f:
FileNotFoundError: [Errno 2] No such file or directory: 'trained_model
    /tab/annotations/instances_val2014.json'
```

Step-by-step process for JSON generation is detailed at tableimg_to_xml repository, as learnt from the first author of (3).

### 2.3 Memory issues

With tensorflow-gpu on a laptop with Nvidia 1050 TI 4GB, I kept running into memory allocation issue, with no output. CPU-based script run generated jpg results output. Reducing hyperparameters, e.g., reducing 'DETECTION_MAX_INSTANCES' seems to help.

### 2.4 The current script leads to 'TypeError', minor fix necessary

With a CPU-based run, while one does manage to get the jpg result with the detected cells (cf. Figure 3), one gets a TypeError.

```
Traceback (most recent call last):
File "samples/tabnet/tabnet.py", line 592, in <module>
limit=int(args.limit))
File "samples/tabnet/tabnet.py", line 404, in evaluate_tabnet
tab_results = tab.loadRes(results)
File "/home/<user>/anaconda2/envs/tf1/lib/python3.7/site-packages/
    pycocotools/coco.py", line 325, in loadRes
annsImgIds = [ann['image_id'] for ann in anns]
File "/home/<user>/anaconda2/envs/tf1/lib/python3.7/site-packages/
    pycocotools/coco.py", line 325, in <listcomp>
annsImgIds = [ann['image_id'] for ann in anns]
TypeError: list indices must be integers or slices, not str
```

Changing 'tab_results = tab.loadRes(results)' to 'tab_results = tab.loadRes(results[0])' fixes the problem of running into the TypeError above, and of the consequent early termination of the program.

| Model | Dimension | Feature Representations | Mean CCC | Std. Dev. CCC |
|---|---|---|---|---|
| SVR | Arousal | LLDs | .122 | .106 |
| | | Functionals | .232 | .164 |
| | | Bag-of-LLDs | **.327** | .208 |
| | | Bag-of-Functionals | .178 | .126 |
| | Valence | LLDs | .055 | .094 |
| | | Functionals | .123 | .137 |
| | | Bag-of-LLDs | **.162** | .184 |
| | | Bag-of-Functionals | .067 | .819 |
| GRU-RNN | Arousal | LLDs | .189 | .227 |
| | | Functionals | .143 | .284 |
| | | Bag-of-LLDs | **.370** | .237 |
| | | Bag-of-Functionals | .328 | .203 |
| | Valence | LLDs | .136 | .175 |
| | | Functionals | .213 | .235 |
| | | Bag-of-LLDs | .191 | .216 |
| | | Bag-of-Functionals | **.223** | .201 |
| Unweighted Average | Arousal | LLDs | .192 | .202 |
| | | Functionals | .204 | .220 |
| | | Bag-of-LLDs | **.370** | .229 |
| | | Bag-of-Func. | .292 | .179 |
| | Valence | LLDs | .130 | .134 |
| | | Functionals | .210 | .204 |
| | | Bag-of-LLDs | **.218** | .204 |
| | | Bag-of-Func. | .176 | .151 |
| Weighted Sum | Arousal | LLDs | .196 | .225 |
| | | Functionals | .185 | .244 |
| | | Bag-of-LLDs | **.372** | .231 |
| | | Bag-of-Func. | .311 | .190 |
| | Valence | LLDs | .138 | .165 |
| | | Functionals | **.219** | .224 |
| | | Bag-of-LLDs | .216 | .212 |
| | | Bag-of-Func. | .203 | .181 |

(a) Table 1

(b) Table 1 predictions

| | | Cultures | | | | | | Total |
|---|---|---|---|---|---|---|---|---|
| | | ZH | EN | DE | GR | HU | SB | |
| Gender | Male | 36 | 33 | 39 | 34 | 26 | 33 | 201 |
| | Female | 34 | 33 | 25 | 22 | 44 | 39 | 199 |
| Interactions | F-F | 22 | 20 | 16 | 8 | 30 | 16 | 112 |
| | M-F | 24 | 26 | 18 | 26 | 28 | 46 | 168 |
| | M-M | 24 | 20 | 30 | 22 | 12 | 10 | 118 |
| Age | 18–29 | 44 | 34 | 41 | 18 | 44 | 22 | 203 |
| | 30–39 | 16 | 12 | 13 | 29 | 9 | 15 | 94 |
| | 40–49 | 4 | 6 | 1 | 1 | 5 | 8 | 25 |
| | 50–59 | 6 | 8 | 5 | 8 | 5 | 14 | 46 |
| | 60+ | 0 | 6 | 4 | 0 | 7 | 13 | 30 |
| Total | | 70 | 66 | 64 | 56 | 70 | 72 | 398 |

(c) Table 2

(d) Table 2 predictions

Figure 3: Table images provided as the test inputs to the model, and the corresponding outputs. The input images were zero-padded to approximately 8-times their original size going by the default configuration of the model. The white spaces from the images in the second column above are cropped out to make the output image look the same size as the input for an easy comparison.

## 2.5 XML generation module

Step-by-step process for JSON generation is detailed at tableimg_to_xml repository, as learnt from the first author of (3).

# 3 Results

For the test images provided as an input, the outputs were obtained as shown in Figure 3. We notice that the table cells have all been mostly correctly identified.

## 3.1 Image output

We also note that, there is likely a fix necessary in the pre-processing module, since both the images have been observed to expand to a 1600x1600 pixel square, with zero padding. The two input image sizes were 772x422 pixels and 407x560 pixels. While the default 'IMAGE_RESIZE_MODE' is 'square', the config.py clearly states the following:

```
        In this mode, images are scaled up such that the
        small side is = IMAGE_MIN_DIM, but ensuring that the
        scaling doesn't make the long side > IMAGE_MAX_DIM.
```

Interestingly, as per the config file, IMAGE_MIN_DIM = 800 and IMAGE_MAX_DIM = 1024, both less than 1600, contrary to what is said above. Unnecessarily high dimensions of the input image make the predictions memory intensive, i. e., computationally expensive.

## 3.2 TXT output

There is also a text output of the following form.

```
tablecell 0.9993474 682 370 771 416
tablecell 0.9990779 334 369 406 416
tablecell 0.99901736 0 74 159 139
tablecell 0.99894994 543 369 607 416
tablecell 0.998898 258 370 335 414
```

## 3.3 XML output

The xml outputs obtained were of the following form.

```
<?xml version="1.0" encoding="UTF-8"?>
<prediction>
<folder>images</folder>
<filename>input_images_SewaDemoTable</filename>
<path>gt_without_box/input_images_SewaDemoTable.
    jpginput_images_SewaDemoTable</path>
<source>
<database>Unknown</database>
</source>
<size>
<width>772</width>  <height>422</height>  <depth>3</depth>
</size>
<segmentated>0</segmentated>
<object>
<name>table</name>
<pose>Unspecified</pose>
<truncated>0</truncated>
<difficult>0</difficult>
<bndbox>
<xmin>0</xmin> <ymin>0</ymin> <xmax>772</xmax> <ymax>422</ymax>
</bndbox>
<cells>
<tablecell>
<dont_care>False</dont_care>
<end_col>11</end_col>    <end_row>16</end_row>
<start_col>11</start_col>  <start_row>11</start_row>
<x0>682</x0>  <x1>771</x1>  <y0>370</y0>  <y1>416</y1>
</tablecell>
<tablecell>
<dont_care>False</dont_care>
<end_col>4</end_col>    <end_row>16</end_row>
<start_col>4</start_col>    <start_row>4</start_row>
<x0>334</x0>  <x1>406</x1>  <y0>369</y0>  <y1>416</y1>
</tablecell>
...
</tablecell>
</cells>
</object>
</prediction>
```

We note that the XML output does not feature the contents of the cells, but it does feature the row and column ids, their start and end coordinate locations, with quite a high confidence score as listed in the text output copied above.

## 4  Conclusion

The TabStructNet was found to be quite good at detecting the locations of the cells in a table. JSON and XML generation module requires manual intervention. Some inconsistencies between the description in the paper and the implementation

were discovered and reported. The process can be streamlined by invoking the corresponding scripts and the necessary file copying operations, as part of the main script itself. There is an unusual image expansion happening which might be further responsible for increasing the memory requiements. It is hoped that this report would help an average user as well as the authors of TabStructNet identify the core issues with the shared repository, fixing which would help raise the popularity of the model in the upcoming ICDAR challenges. Incremental improvements should be particularly interesting and to be look forward to, in making the model faster, more accirate and precise.

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
