# OpenReview forum: "How effective is TabStructNet in capturing the structure of a table-image into an XML? : A reproducibility report"
_ML_Reproducibility_Challenge/2020 — Reject_

### Official Review · AnonReviewer3 · 2021-02-28
**Good code reproduction, but no numerical results**

**Rating:** 3
**Confidence:** 4

**Review:**

The submission did a good job in communicating with the authors, and because of that it has identified a missing component (a JSON meta-data generator) and a discrepancy between code and paper on the neural network design. This enabled the submission to reproduce the running pipeline of the model mostly based on the original authors' public code.

However, the submission did not report any numerical results that can be used to judge the reproducibility of the original paper. It only quantitatively states that the produced results seem good with high confidence scores. The purpose of the reproducibility is to verify such numerical results, therefore the submission is a clear rejection.

That said, I believe if given more time, the submission could become a good reproduction of the original paper with clear numerical results, and perhaps even some additional ablation study.

**Familiar With The Original Paper:**

I have read the original paper

**Reproducibility Summary:**

Report has summary

---

### Official Review · AnonReviewer2 · 2021-03-01
**Review on Paper 87**

**Rating:** 4
**Confidence:** 3

**Review:**

This reproducibility report is about TabStructNet which was published in ECCV 2020.

Authors of this report provide summary of report, scope of reproducibility and communicated with original author of the TabStructNet.


Basically, there are available codes from the original authors, so this report tried to discover missing points in the original paper. Also, they experimented with missing modules such as the post-processing module, JSON generation module, and FPN. Therefore, it is important to perform plenty of “hyperparameter search” and “ablation” to analyze them.

However, ablation studies and hyperparameter search are very limited in this report. Also, discussion on results is not enough. They just provide several examples and short conclusions.


Line4, there is a typo: EECV 2020 → ECCV 2020


**Familiar With The Original Paper:**

I have not read the original paper

**Reproducibility Summary:**

Report has summary

---

### Official Review · AnonReviewer1 · 2021-03-02
**The submission provides useful insights into running the existing code repository, but fails to conduct a representative experimental study.**

**Rating:** 4
**Confidence:** 4

**Review:**

The paper reproduces TabStructNet, an approach to automatically translate tables to XML format. The authors of the submission make use the provided code repository of the original authors and document the required changes and extensions to get the code running as well as available divergences of the implementation with respect to the paper.

However, the paper does not reproduce a representative set of results of the original paper, as only exemplary inputs and outputs of the model are presented. While it provides useful information to the reader, as the available code repository is analysed in terms of functional reusability, the paper misses to analyse the quality of the implementation on original or representative datasets with original metrics. There is also no ablation study present in the submission.

**Familiar With The Original Paper:**

I have read the original paper

**Reproducibility Summary:**

Report has summary

---

### Decision · Program_Chairs · 2021-03-31

**Decision:**

Reject

**Comment:**

Overall reviews and/or the paper content not good enough for the AC to recommend to the journal.